# From Single- to Multi-Family Public Housing: Analyzing Social Sustainability Aspects of Recent Designs in the UAE

**Omar Sherzad M.Shareef and Khaled Galal Ahmed \***

Architecture Engineering Department, College of Engineering, United Arab Emirates University,
Al Ain P.O. Box 15551, United Arab Emirates; 700040309@uaeu.ac.ae
\* Correspondence: kgahmed@uaeu.ac.ae

**Abstract:** The UAE's federal and local governments initiated their public housing programs for Emirati citizens on low incomes to provide them with adequate basic services and to improve their living conditions. Until 2005, most of the public housing units were developed as single-family dwellings on ample plots, but, afterwards, the areas of the housing plots significantly decreased due to the growing demand for public housing associated with the increasing population and limited land availability. Recently, it has become increasingly difficult to provide single-family housing for all Emirati citizens who need public dwellings. To address this problem, the UAE has shifted towards the provision of multi-family housing. To this end, two pioneering vertical public housing projects have been developed: Diba Al Hosn in Sharjah and Al Ghurfa in Al Fujairah. While the designs of the two projects attempted to consider the requirements of the lifestyle of Emirati families, the degree of attaining the wider social sustainability considerations in the design of these projects has not been explored yet. To bridge this gap, this research first examined the essential social sustainability aspects that should be taken into consideration when designing multi-family housing in general. Then, the research compared the two multi-family local designs with some global case studies regarding the degree of achieving social sustainability aspects. The research ended with revealing the social sustainability-related shortcomings of the designs of recent multi-family public housing in the UAE and, hence, suggested an integrated set of social sustainability principles and indicators that, if appropriately applied, could help appropriately attain social sustainability aspects for Emirati's vertical public housing.

**Keywords:** Emirati citizens; multi-family housing; social sustainability; public housing; UAE

## 1. Introduction

Starting in the early 1970s, the United Arab Emirates (UAE) initiated public housing programs at both the federal and local government levels with the aim of providing affordable housing to Emirati citizens on low incomes. These programs were designed to guarantee that everyone has equal access to essential services and to enhance the overall living conditions of the UAE residents (ADHA 2016). The primary federal organization responsible for providing public housing in all seven Emirates is the Sheikh Zayed Housing Program (SZHP), which was founded in 1999 (Sheikh Zayed Housing Program 2020). Since then, each Emirate has established its own local public housing agency to increase the availability of housing for Emirati citizens. These local programs include the Mohammed Bin Rashid Housing Establishment (MRHE) in Dubai (MBRHE Mohammed Bin Rashid Housing Establishment (MRHE 2021)), the Sharjah Housing Program in Sharjah (Sharjah Housing Program 2022), the Abu Dhabi Housing Authority (ADHA) in Abu Dhabi (ADHA 2023), and the Sheikh Saud Housing Program in Ras Al Khaimah (The Sheikh Saud Housing Program 2019).

All these public housing agencies offer various sizes and designs of single-family housing units to low-income Emirati citizens (Galal Ahmed 2017). But generally, most of

the designs of the public housing units used to be comprised of two distinct buildings on the same plot with areas ranging from 2000 m$^2$ to 3000 m$^2$. The first building, situated at the front, consisted of a single floor allocated for a male guest hall and a dining room with associated services. The second building was designated for family activities and a kitchen (Galal Ahmed 2021; Galal Ahmed et al. 2022). With its careful attention to the spatial functional relationship among the different habitable spaces of the houses, this typical design also used to cater to the socio-cultural aspects of Emirati families, especially privacy (Galal Ahmed 2011). However, after 2005, the public housing plot area decreased to approximately 750 m$^2$ or even less to respond to the need to develop more houses to satisfy the increasing demand due to population growth and limited land resources (Galal Ahmed et al. 2022). Still, even with such an action, it remained increasingly challenging to provide affordable single-family housing for all Emirati citizens in need of public housing. To tackle this issue, both local and federal governments in the UAE have shifted their focus towards developing affordable multi-family housing designed for Emirati citizens in recent years. This shift might be attributed to both economic and sustainable development reasons. Economically, developing mid- and high-rise residential blocks would help provide larger numbers of social housing dwellings compared to single-family housing. As for sustainable development, such types of vertical social housing help preserve land and conserve resources, especially related to infrastructure and roads (Galal Ahmed 2012).

Moreover, achieving the United Nations' 17 Sustainable Development Goals (SDGs), especially Goal 11, Sustainable Cities and Communities (making cities inclusive, safe, resilient, and sustainable), and Goal 13, Climate Change Action (taking urgent action to combat climate change and its impacts), was another motivation towards this shift in public housing typology (United Arab Emirates Government 2023). According to the adopted SDG agenda by the federal and local governments in the UAE, all sectors of urban development including public housing should comply with this agenda to ultimately achieve the defined SDGs by 2030 (National Committee on Sustainable Development Goals 2017). From a global perspective, it is also believed that vertical housing communities are considered the most viable and environmentally friendly living option for the future. Furthermore, multi-family housing has enhanced proximity among diverse families and individuals (Alshutti 2021).

In recent years, the UAE's federal and local government has developed two vertical public housing projects. The first is the Diba Al Hosn project in Sharjah and the second is the Al Ghurfa Project in Al Fujairah. As officially claimed, these two projects aimed to offer a high-quality living environment for their Emirati residents, minimize the use of green field lands, and reduce the waiting time for acquiring public housing units for Emiratis (Albayan 2021; EN Agency 2023). Until now, it is not evidently clear to what extent the designs of these pioneering vertical public housing projects have managed to actually satisfy social sustainability aspects by providing a better quality of life for Emirati citizens who live in such multi-family buildings. To bridge this gap, the research set some objectives to, first, develop a conceptual framework for social sustainability aspects in vertical housing, specifically, and second, to utilize this conceptual framework to assess the designs of the two Emirati vertical public housing projects. Accordingly, the research poses the following two questions: What are the key aspects that contribute to social sustainability in vertical housing? And do the two recently developed vertical public housing projects in the UAE comply with the developed conceptual framework for social sustainability?

This study ultimately hopes to shed some light on the social sustainability aspects of this new public housing typology, especially from a social sustainability point of view. So, it contributes to the efforts of achieving the SDGs as per the adopted local agendas in the UAE. On the other hand, the study contributes to the current debate about achieving social sustainability in vertical public housing where many efforts in this regard have been discussed globally but with very little attention paid to such an experience in this part of the world.

## 2. Methodology

The research methodology encompasses two phases to address the research questions (Figure 1). In the first phase, a conceptual framework for social sustainability aspects was developed based on, first, a review of relevant literature about the main principles for social sustainability in vertical housing, including functional mixed-use development, social interaction, accessibility, security, privacy, high-quality living environments, user needs, and community involvement in design. And second, the analysis of some case studies in both developed and developing countries including the Netherlands, United Kingdom, Denmark, France, Singapore, China, and Mexico. These global case studies were selected based on their relevance to the social sustainability aspects of multi-family housing. This conceptual framework has provided a comprehensive understanding of the social sustainability aspects that need to be considered in the study context of the research.

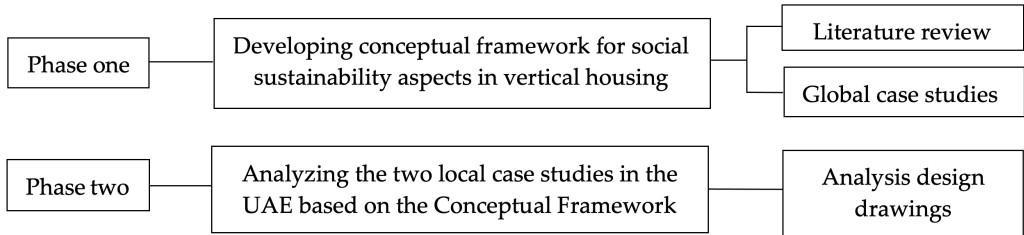

**Figure 1.** The two-phase research methodology to answer the research questions.

In the second phase of the research analysis, the developed conceptual framework was utilized to examine the designs of the two vertical public housing projects in the UAE; namely, the Diba Al Hosn project in Sharjah and the Al Ghurfa project in Al Fujairah. This analysis was conducted using the various relevant design drawings and photographs, such as the floor plans, the building elevations, and the 3D perspectives, to determine the extent to which the designs of these projects have considered the principles of social sustainability.

## 3. The Developed Conceptual Framework

According to the studies conducted by Bojago (2022) and Shehab and Kandar (2021), social sustainability can be defined as the measurement of human well-being and the improvement of well-being for both present and future generations. Therefore, it emphasizes the consideration of human needs during the design of housing, which ultimately enhances the quality of the built environment. Key aspects of social sustainability include social interactions and accessibility, social integration and security, functional mixed-use, and privacy. In relation to urban development, researchers such as Mohamed et al. (2022) and Abdulrahman and Motlak (2023) have highlighted the importance of mixed-use spaces, urban patterns, and community participation. Consequently, social sustainability can be identified as a theme that encompasses community participation, privacy, security, social equity, and social interactions. As mentioned earlier, the conceptual framework for addressing social sustainability aspects in vertical housing has been developed through a relevant literature review, focusing on these main principles in the literature besides examining eight case studies from various developing and developed countries around the world.

### 3.1. Literature Review for Social Sustainability Aspects
3.1.1. Functional Mixed-Use

Functional mixed-use is a strategy for sustainable urban growth in densely populated metropolitan areas. By incorporating various types of land use within the same district, such as residential and commercial zones, functional mixed-use development offers benefits like improved property values, energy efficiency, reduced traffic congestion, increased satisfaction among residents, the integration of public uses, and increased municipal revenues (Zhong and Hui 2021). This approach creates comprehensive residential areas that enhance accessibility and attractiveness for both retail and residential properties

(Abdelsalam 2018). Modern urban design emphasizes the principle of mixed-use, replacing strict zoning with multifunctional high-rise buildings that serve as integrated vertical cities, providing high-density and high-quality buildings to support convenient city living and urban densification (Generalova and Generalov 2020).

### 3.1.2. Social Interaction

Social interaction plays a significant role in achieving social sustainability (Abdullah and Ahmad 2022). The presence of appropriate social spaces for interaction in residential areas is crucial for encouraging social engagement, promoting community development, improving well-being, and enhancing happiness (Nguyen et al. 2020). These social spaces facilitate social contact and relationships among neighbors, leading to greater overall housing satisfaction. The importance of social interaction on the local community level is vital for improving the overall quality of social life and fostering a strong sense of place attachment. Overall, social interaction is essential for individuals' emotional well-being and is a vital component of residential environments as it helps foster a sense of identification with the place (Yao 2020).

### 3.1.3. Accessibility

Accessibility involves providing convenient entry points and vertical movement options, including stairs, ramps, lifts, escalators, and travellators, with the aim of enabling all users, including those with physical disabilities, to perform necessary activities and functions within the residential building. According to Wie and Dewi (2019), "vertical" accessibility connections to shared public spaces in residential buildings can be used to enhance social encounters among residents, allowing them to overcome social separation and increase their sense of security and belonging. Easily accessible public spaces inside vertical residential buildings serve as platforms for artistic expression and displaying power and resistance among residents (Astarini and Utomo 2020). In addition, this emphasizes the importance of considering accessibility in vertical residential building design to ensure easy access to public facilities such as eateries, worship places, parking, and sports facilities within the multi-family block (Sholanke et al. 2020).

### 3.1.4. Security

Security in high-rise residential buildings is a crucial measure that aims at ensuring the safety of residents and property in these buildings. It involves a range of procedures to prevent, minimize, and respond to potential dangers and emergencies caused by natural or human-made threats. Security professionals in high-rise residential buildings have various responsibilities, such as managing the lobby, monitoring CCTV systems, screening visitors, conducting building inspections, and controlling access to the premises (Lee 2019).

### 3.1.5. Privacy

The notion of privacy can be understood by analyzing the layout of internal spaces and perceiving how similar functions are organized into distinct zones based on a hierarchy of privacy for activities and users. For instance, in residential units, there are typically three types of zones: the private zone for specific individuals like the master bedroom and children's bedrooms, the semi-private zone used by all family members such as the living room and kitchen, and the public zone primarily used for entertaining guests. These zones can be identified using the territoriality concept, where the private zone represents the primary territory exclusively used by individual occupants, the semi-private zone serves as the secondary territory used both individually and collectively, and the public zone acts as the tertiary shared space (Tomah et al. 2016).

### 3.1.6. High Quality of Living Environment

To create a high-quality living environment for apartment residents, various factors need to be considered. These include ensuring access to natural light, sunlight, and

different views, which promote residents' health and well-being. It is also important to focus on thermal and acoustic comfort, as well as indoor air quality. To prevent overheating, especially in a changing climate, measures such as summer solar shading, vegetation, and natural night-time ventilation can be implemented. Key design principles involve the use of shallow-plan buildings and layouts, multiple windows in different directions to maximize daylight and outdoor connections, and the incorporation of balconies that are spatially connected to the main living spaces (Astarini and Utomo 2020)

### 3.1.7. User Needs

Recent studies have emphasized the significance of housing design in meeting residents' needs (Itma 2018). Recognized in various international human rights instruments, the right to adequate housing goes beyond mere shelter and encompasses a wide range of complex human desires and requirements. The relationship between housing design and residents' needs is crucial and should be comprehensively examined. Maslow's theory of needs suggests that individuals have a hierarchy of needs, with higher needs emerging once lower needs are fulfilled. This hierarchy includes physiological needs, security, social affiliation, self-esteem, self-realization, knowledge, and aesthetic needs. Understanding and addressing these needs in the design of vertical residential buildings are essential for residents' satisfaction (Ahmed Kamer Eldawla 2022).

### 3.1.8. Residents' Involvement in the Design

Residents' participation is crucial for sustainable housing development, as it promotes sustainability and the creation of suitable design solutions. When it is conducted as a participatory process, the design of vertical residential buildings allows both designers and residents to express their thoughts and reach satisfactory design decisions. Such a participatory process should occur at various stages, such as planning, design, construction, and post-occupancy evaluation. When performed adequately, this participatory process helps prevent unnecessary modifications after occupancy (Moghimi et al. 2017). Residents' participation allows the redistribution of decision-making power to include those typically excluded, fostering communication and encouraging healthy collaboration between residents and designs. So, the participatory process in itself is inherently educational, because it combines the expertise of professional designs with the diverse life experiences of residents, resulting in a unique and enriched final housing design (Fakere et al. 2017).

Table 1 summarizes the aforementioned aspects of social sustainability in multi-family residential buildings.

**Table 1.** The definition of the social sustainability aspects based on the literature review.

| Social Sustainability Aspects | Description |
| --- | --- |
| Functional mixed-use | Functional mixed-use is an essential option for achieving social sustainability by incorporating a combination of housing, commercial zones, and residential areas in the same location. |
| Social interaction | Social interaction plays a significant role in residents' daily lives, well-being, and happiness. |
| Accessibility | In vertical housing, accessibility should be prioritized to ensure easy access to shared facilities such as shops, public spaces, car parking, and sports facilities. |

**Table 1.** *Cont.*

| Social Sustainability Aspects | Description |
|---|---|
| Security | Security measures in multi-family housing should include a lobby concierge, a monitoring CCTV camera system, and access control. |
| Privacy | When designing multi-family housing, privacy should be considered for both the individual apartment units and the overall project, incorporating private zones, semi-private zones, and public spaces. |
| High quality of living environment | This can be achieved through access to daylight and ventilation, the inclusion of shrubs and trees, and the use of sustainable materials. |
| User needs | The design should prioritize meeting the users' satisfaction by considering the importance of their needs. |
| Residents' involvement in the design | Residents' participation in the design process is crucial to achieving sustainable housing, and technologies such as virtual reality can be utilized to involve the residents effectively. |

*3.2. Social Sustainability Aspects Based on the Global Case Studies*

In this section, eight case studies for the designs of vertical residential buildings in both developed and developing countries are analyzed based on the main aspects of social sustainability in multi-family housing, as defined in the literature review. The main objective is to enrich the conceptual framework through an understanding of how these projects managed to implement some or all social sustainability aspects in their designs. Tables 2 and 3 conclude the defined implementation methods in the designs of these buildings.

**Table 2.** Social sustainability principles for vertical housing case studies in developed countries.

| Social Sustainability Aspects | Trudo Vertical Forest Housing in the Netherlands (Stefano Boeri Architetti 2023) | TR Hamzah and Yeang's Approach, "City-in-the-Sky in London, UK" (Yeang 2007) | Eight-House Bow-Shaped Building in Denmark (Gonchar 2011) | LA SERRE D'ISSY Residential Housing in France (MVRDV | La Serre 2017) |
|---|---|---|---|---|
| Functional mixed-use | A residential tower with mixed-use on the ground and first floor, combining residential and commercial spaces. | A mixed-use development featuring a variety of apartment types. | Different types of housing units with commercial units on the ground floor. | A variety of residential units with a commercial zone on the ground floor. |
| Social interaction | Vertical forests: Each apartment is equipped with a balcony, allowing residents to experience the feeling of living in a house. | The ground floor features open spaces with public facilities, along with landscaped areas within each residential tower. | The design includes two courtyards in the center, arranged in a bowtie shape, to foster increased social interaction among residents. | Gardens and terraces are incorporated among the residential apartment floors, enhancing social life and leisure opportunities. |
| Accessibility | Easy access to public spaces and proximity to the city center. | Direct connection to open spaces on every floor and proximity to the commercial zone in the city. | Open ramp for easy access to the townhouses and penthouses, and it is situated near the lake. | Open staircases connecting each floor to encourage residents to socialize with each other within the building. |
| Security | The entrance lobby features a 24 h concierge and CCTV camera system. | Each residential tower has a 24 h concierge in the lobby entrance. | The residential units have a private entrance and CCTV camera system. | The entrance lobby offers a 24 h concierge service and is equipped with a CCTV camera system. |

**Table 2.** *Cont.*

| Social Sustainability Aspects | Trudo Vertical Forest Housing in the Netherlands (Stefano Boeri Architetti 2023) | TR Hamzah and Yeang's Approach, "City-in-the-Sky in London, UK" (Yeang 2007) | Eight-House Bow-Shaped Building in Denmark (Gonchar 2011) | LA SERRE D'ISSY Residential Housing in France (MVRDV | La Serre 2017) |
|---|---|---|---|---|
| Privacy | The residential tower has a private entrance, and each apartment features a balcony with shrubs and trees. | The design includes a range of spatial public open spaces, progressing from parks in the sky, to semi-private parks, and finally to private open spaces on the balconies of each apartment. | Two interior courtyards have been incorporated to enhance privacy for the residents. | A vegetation structure covers the entire building block, further increasing privacy for the residents. |
| High quality of living environment | Natural ventilation is achieved through the presence of shrubs and trees on the vertical forest balcony. | A healthy landscaped environment is created on balconies, promoting natural ventilation and natural lighting. | The residential units are designed with two interior courtyards and roof gardens, allowing for the entry of natural light and air. | The entire block is covered with a vegetation structure, creating a natural atmosphere for the residents. |
| User needs | Each apartment is equipped with a vertical forest balcony, allowing residents to feel connected with nature. | Every apartment features a garden balcony, providing residents with a connection to nature. | The housing development offers a variety of housing units, including apartments, penthouses, and townhouses. | Each floor of the residential apartments has terraces and gardens, providing space for planting and leisure activities. |

**Table 3.** Social sustainability principles for vertical housing case studies in developing countries.

| Social Sustainability Aspects | Yew Tee's Vertical Kampung Housing in Singapore (Lin 2021) | Baiziwan Social Housing in China (Iype 2022) | Meir Lobaton + Kristjan Donaldson—Residential in Mexico (Cilento 2009) | Linked Hybrid in Beijing, China (Wilkinson 2016) |
|---|---|---|---|---|
| Functional mixed-use | The residential block and commercial block include a community plaza, retail shops, and an outdoor public area. | Different types of apartment units are available, and the street level includes a range of commercial units. | The residential tower has a single apartment with a private garden on every floor. | The mixed-use complex consists of residential units, a commercial zone, and green open spaces. |
| Social interaction | The development features gardens on multiple levels, children's playgrounds, and a commercial zone. | The first level of the project includes floating gardens and open spaces specifically designed for the residents. | Each apartment is equipped with a residential garden, creating a social area for family members to enjoy. | The project also includes open public spaces such as parks, sitting areas, and fountains, providing opportunities for social interaction. |
| Accessibility | Common corridors and open staircases encourage residents to socialize with each other. | A pedestrian circuit around all blocks enhances connectivity and connects the neighborhood with the city. | The residential block is situated close to public spaces and the city. | Sky bridges are located between upper floors, allowing residents to walk around the complex. |

**Table 3.** *Cont.*

| Social Sustainability Aspects | Yew Tee's Vertical Kampung Housing in Singapore (Lin 2021) | Baiziwan Social Housing in China (Iype 2022) | Meir Lobaton + Kristjan Donaldson—Residential in Mexico (Cilento 2009) | Linked Hybrid in Beijing, China (Wilkinson 2016) |
|---|---|---|---|---|
| Security | Each tower has a 24 h concierge in the entrance and a CCTV camera system. | There is a 24 h concierge in each tower's lobby and a CCTV camera system. | There is a CCTV camera system in place. | Each tower's lobby features a 24 h concierge service. |
| Privacy | Each tower has a private entrance and elevators. | Each tower has a private entrance with landscaped areas and open spaces exclusively for residents. | Each floor has a single apartment with a private garden for family members. | There are well-defined private residential blocks and public-use zones. |
| High quality of living environment | The building incorporates multiple levels of gardens and utilizes sustainable materials to lower energy consumption. | The first floor features landscaped parks, while trees are planted on the ground floor, making use of sustainable materials. | Each apartment is equipped with a private garden, allowing for natural lighting and ventilation through the windows. | The complex boasts landscaped parks, lakes, and natural ventilation facilitated by the open central court. |
| User needs | Designed according to the users' needs, such as a residential block with a commercial block, gardens, and green open spaces. | Designed with a communal outdoor landscape for residents on the first floor and a shopping street on the ground floor. | Each apartment is equipped with a garden, allowing residents to feel connected with nature and providing an attractive area for family members. | Designed to meet the users' needs, including a commercial zone, residential units, and educational and recreational facilities in one place. |

## 4. Results for Analyzing Social Sustainability Attainment in the Two Pioneering Vertical Public Housing Projects in the UAE

### 4.1. Diba Al Hosn Project in Sharjah

The development of the Diba Al Hosn project in Sharjah was initiated in 2014 by the Sharjah Housing Program (Figure 2). This project was envisaged as a representation of an important milestone in the development of socially and environmentally sustainable and affordable vertical public housing designs for Emirati citizens on low income. Each of the residential blocks accommodates 12 apartments in each of the seven typical floors. As shown in Figure 3, each apartment has a total area of approximately 315 m$^2$. On the ground floor, there are two entrances, a waiting area, a few shops, car parking, and a service room. To ensure privacy, each residential unit is divided into three zones. The first zone is for guests (majlis), the second zone is for the family, and the third zone is designated for services (Galal Ahmed 2012; Albayan 2021).

The project design also adopts residential functional use in all its typical floors with some limited public uses on the ground floor (reception, parking, shops, and technical services). This means that there are limited recreational spaces incorporated into the design, which would negatively affect the quality of the living environment. This vertical public housing project is designed to meet the social and functional needs of Emirati families through the provision of three separate zones within each apartment. Namely, the men's zone contains a guest room (majlis), a dining room, and WC, the family zone includes a living room with a balcony, four bedrooms with bathrooms, one master bedroom, and the service zone. It might be argued that most of the aforementioned social sustainability limitations refer to the focus on economic efficiency and the desire to achieve sustainability through conserving the use of construction and infrastructure resources.

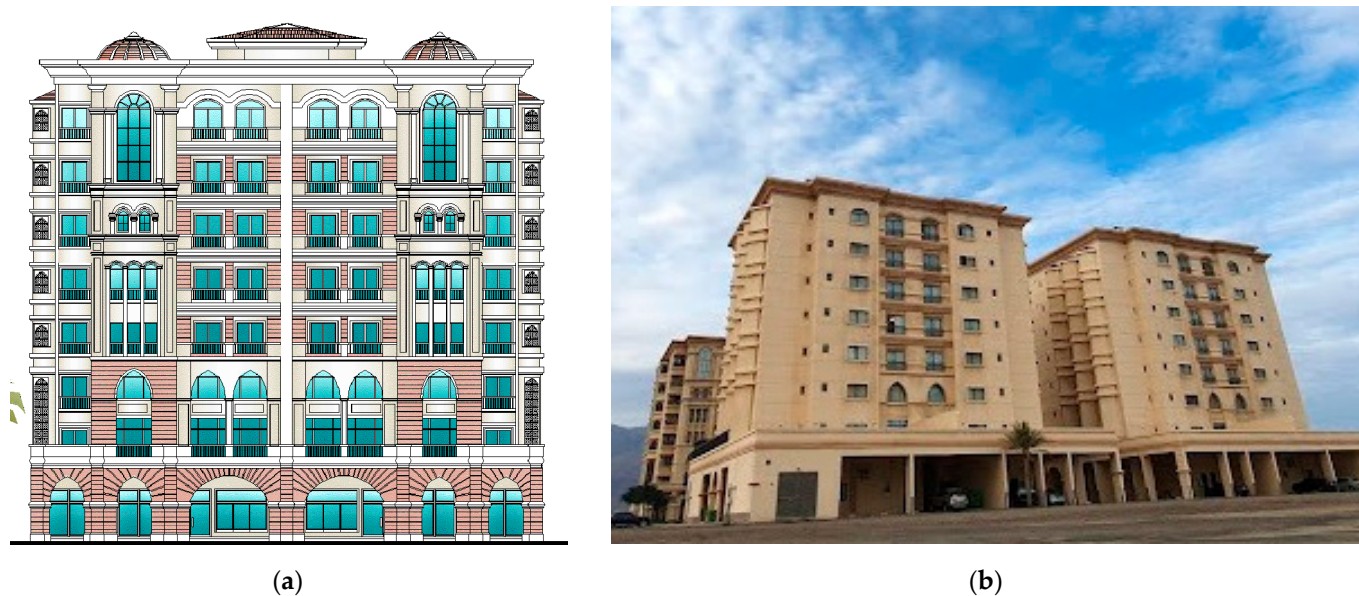

**Figure 2.** Elevation for the Diba Al Hosn multi-family block (**a**) and the as-built view of the blocks (**b**).

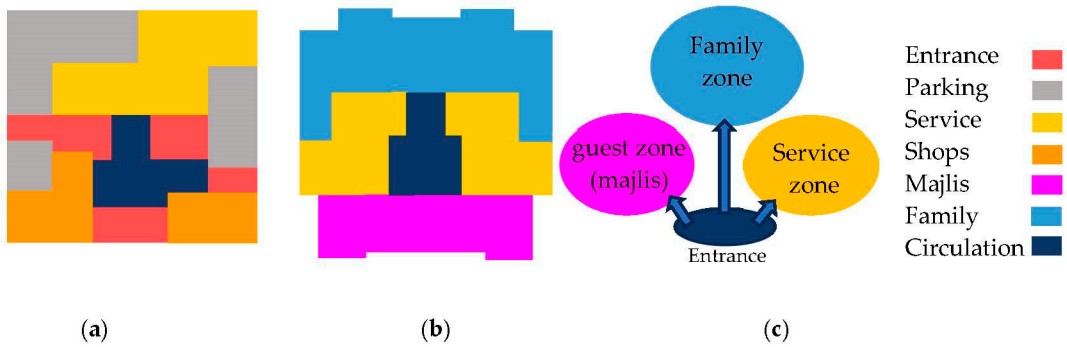

(**a**)          (**b**)          (**c**)

**Figure 3.** Ground floor functional spaces for Diba Al Hosn project (**a**), typical floor functional spaces (**b**), and zoning (**c**).

### 4.2. Al Ghurfa High-Rise Social Housing Project in Al Fujairah

The development of the Al Ghurfa high-rise public housing project was initiated in 2014 by the Sheikh Zayed Housing Program (SZHP) in Al Fujairah Emirate (Figure 4). This public housing project aimed to meet the housing needs of citizens in the Emirate of Al Fujairah by offering them suitable housing accommodations while avoiding long waiting times for obtaining their housing units. The tower comprises 12 vertically arranged "houses", each with an area of about 300m$^2$ (Figure 5). The ground floor of the tower is designated for entrances and car parking. The design of the project is characterized by a unique concept, wherein each flat (house) is divided into three levels. The entrance level (majlis) features a male guest room, a dining room, a kitchen, a storage area, and a guest toilet. The upper level includes a master bedroom, a family living room, a kitchenette, and toilets. The lower level consists of two bedrooms, a maid's room, a laundry room, and toilets (EN Agency 2023; Galal Ahmed 2018).

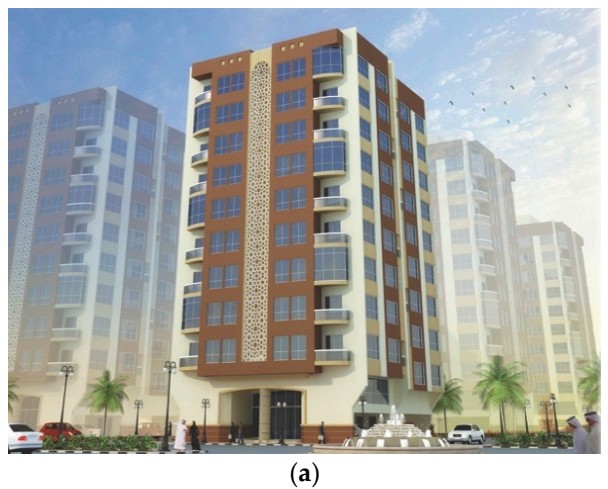
(**a**)

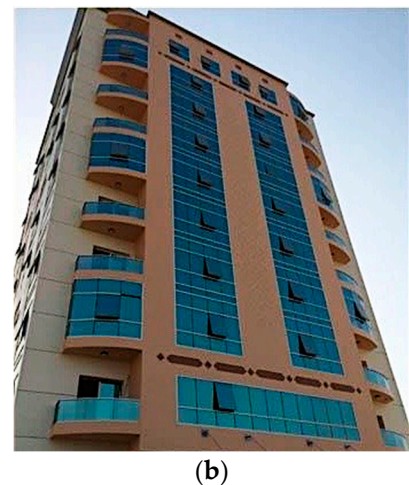
(**b**)

**Figure 4.** Three-dimensional perspective view for the Al Ghurfa high-rise social housing project in Al Fujairah Emirate: as designed (**a**), and as-built view (**b**).

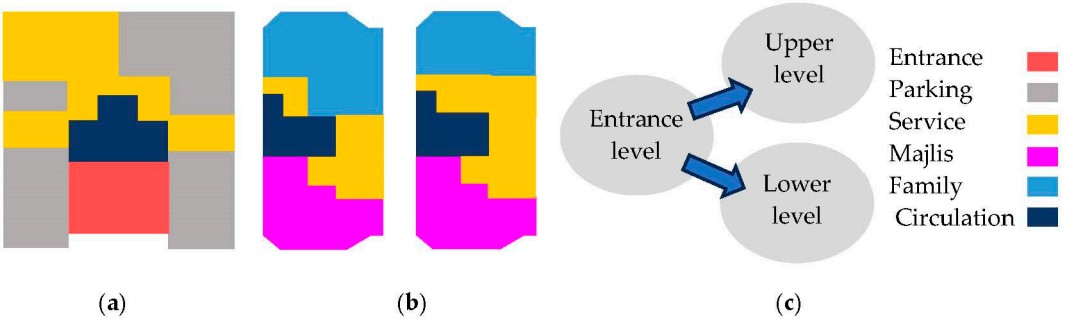
(**a**)                                        (**b**)                                        (**c**)

**Figure 5.** Ground floor functional spaces for the Al Ghurfa high-rise social housing in Al Fujairah Emirate (**a**), typical floor functional spaces (**b**), and the three zoning levels (**c**).

Again, the developed conceptual framework was utilized in the analysis of the design of the Al Ghurfa high-rise social housing project to determine the extent to which it achieves social sustainability aspects. The project is designed as a residential multi-family block with only some limited technical services in the common spaces that allow the possibility for social interaction, except in the main entrance lobby, as shown in Figure 5a. Furthermore, the Al Ghurfa public housing project is situated near the city center, but it still lacks direct access to the surrounding urban area, affecting its overall accessibility. Internally, accessibility is only available through allocated elevators and staircases linking the residential floors with the main entrance. In terms of security, the project incorporates CCTV cameras in the entrance lobby and 24 h concierges to ensure the safety of the building. Concerning privacy, the Al Ghurfa project is designed to meet the privacy requirements of Emirati families, where it features one apartment on each floor, directly accessed by elevators from the main entrance lobby.

However, the project design lacks the incorporation of any recreational features that enhance the quality of the living environment. A unique aspect of the design is the division of each flat (house) into three levels to meet the needs of Emirati citizens. The entrance level of each flat comprises a male guest room (Majlis), a dining room, a kitchen, a storage area, and a toilet. The lower level includes two bedrooms with a shared bathroom, a maid's room, and a washing room. The upper level of the flat consists of the master bedroom, a family living room, and a kitchen (Figure 5b,c). The Al Ghurfa project has been recently constructed and delivered to the occupants but with no direct participation of the residents in the design of the project. Similar to the case of the Diba Al Hosn project, and despite a noticeable enhancement in some social sustainability aspects in Al Ghurfa project, the

limitation of the other social sustainability aspects might be due to reasons related to economic considerations and the reservation of construction resources. Finally, Table 4 summarizes the analysis results of the two projects.

**Table 4.** Summary of the social sustainability analysis of the two pioneering Emirati vertical public housing projects based on the eight principles of social sustainability.

| Social Sustainability Aspects | Diba Al Hosn Project in Sharjah | Al Ghurfa High-Rise Social Housing in Al Fujairah |
|---|---|---|
| Mixed-use | The ground floor contains parking, an entrance lobby, shops, and services | It does not have any mixed-use areas; it solely consists of parking, an entrance lobby, and services. |
| Social interaction | There is no social interaction in the four towers. | This vertical housing does not promote social interaction. |
| Accessibility | It is located on an island, and there is no accessibility to the surrounding areas. | It is situated in a residential district without any accessibility. |
| Security | The main entrance lobby has a CCTV camera system and 24 h concierges for each tower. | The main entrance lobby is equipped with a CCTV camera system and has 24 h concierges for each tower. |
| Privacy | This project is designed according to the privacy of Emirati families. | This project is designed with the privacy of Emirati families in mind. Additionally, each floor has one apartment. |
| High quality of living environment | There are no gardens or natural materials to enhance the quality of the living environment. | There are no trees or natural elements incorporated to improve the living environment's quality. |
| User needs | This vertical housing is composed of three separate zones in each apartment. | This multi-family public housing is composed of three separate levels in each house. |
| Community involvement in the design | Emirati residents have not participated in the design of this vertical housing. | Emirati residents have not participated in the design of this project. |

## 5. Discussion

A few years ago, the UAE federal and local governments shifted their focus towards providing affordable multi-family public housing solutions for mainly young Emirati citizens (mostly at the marriage ages of 20 to 30) who have low incomes. The appropriate design of vertical public housing in the UAE improves the living conditions of Emirati citizens by promoting social sustainability. Moreover, vertical public housing seems to be the most suitable solution for the UAE's local and federal governments to meet the expanding demand for public housing, especially by young Emirati citizens. With the recent development of two pioneering projects for multi-family housing in the UAE, this study assessed through the developed conceptual framework the extent to which each of the two designs managed to achieve the eight defined social sustainability aspects, including mixed-use, social interaction, accessibility, security, privacy, high quality of the living environment, user needs, and community involvement in the design. The analysis of the global case studies in both developed and developing countries demonstrated the followed method of implementing the social sustainability aspect through various design approaches.

The Diba Al Hosn project specifically achieved four out of the eight main defined social sustainability aspects. Firstly, functional mixed-use spaces such as the provision of some shops, cafes, restaurants, and services on the ground floor to serve the residents in the block. Secondly, security was ensured by allocating a private entrance with a lobby

leading to the apartments and a CCTV camera system. Thirdly, privacy was achieved by designing separate entrances and elevators for men, women, or families in each apartment building. Lastly, user needs were addressed by dividing each apartment into three zones, including a guest zone (majlis), a family zone, and a service zone. The remaining aspects of social sustainability were absent in the design.

On the other hand, the Al Ghurfa social housing project achieved three out of the eight principles. Firstly, privacy was emphasized by designing one apartment on each floor with two separate entrances: one for men and another one leading to the lobby. Secondly, security measures were implemented, such as CCTV cameras in the entrance lobby and 24 h concierges to ensure resident safety. Lastly, user needs were considered by dividing each flat (house) into three levels: the entrance level, the upper level, and the lower level, each with appropriately allocated functions. This indicates that the design of the Al Ghurfa project missed five of the main social sustainability aspects.

These results indicate that there is a desperate need to reconsider the design of vertical public housing in the UAE to address the identified gaps, especially concerning social interaction, high quality of living, and residents' participation. These gaps might be bridged by benefiting from the global experiences in both developing and developed countries, as briefly discussed above (see references: Stefano Boeri Architetti 2023; Yeang 2007; Gonchar 2011; MVRDV | La Serre 2017; Lin 2021; Iype 2022; Cilento 2009; Wilkinson 2016). For example, the designs of the multi-family housing blocks could incorporate recreational and social gathering elements such as gardens on multiple vertical levels and/or allocated private gardens for each apartment. The design could also encompass children's playgrounds, open spaces with public facilities on the ground floor, and an open courtyard for the entire residential building to encourage social interaction. Additionally, a mixed-use design combining residential and commercial zones, along with different types of housing units, is essential in attaining a socially sustainable design. Designing for accessibility can be internally enhanced by the inclusion of bridges, ramps, elevators, and open staircases to facilitate easy access to shops, open spaces, courtyards, and parking areas. Also, accessibility should be considered for the supporting services and facilities in the wider urban context.

Furthermore, besides incorporating recreational features such as courtyards and balcony gardens with trees and shrubs in the design, proper ventilation through windows and the use of natural materials can contribute to a high-quality living environment. Lastly, community involvement in the design process can be realized by utilizing traditional and modern participation tools, such as virtual reality technologies. These technologies enable individuals to experience and interact with virtual representations of the proposed housing designs. Through this, Emirati citizens can visualize how the vertical housing units will look and explore various design options as needed. Meanwhile, the appropriately achieved social sustainability aspects of privacy, user needs, and security in the designs of the two analyzed projects of Diba Al Hosn and Al Ghurfa should be preserved and enhanced in future vertical public housing projects in the UAE.

## 6. Conclusions

Attaining social sustainability in the design of the newly introduced public housing typology of vertical residential blocks in the UAE is essential to providing good living conditions for low-income Emirati citizens and promoting overall sustainability development. The conceptual framework for the social sustainability aspects of vertical residential housing has been successfully developed in this research through two resources. First, a literature review was conducted to explore the related principles and indicators. Second, eight global case studies for vertical housing were analyzed to understand the various design approaches for the implementation of the main social sustainability aspects. The success of this conceptual framework in analyzing the two pioneering vertical public housing projects in the UAE helped determine the extent to which these aspects were achieved

in those projects. This makes this conceptual framework applicable as a verification tool to be applied to vertical public housing designs in the UAE.

The revealed obvious lack of the full attainment of social sustainability aspects in the two designs of Diba Al Hosn and Al Ghurfa makes it necessary for the federal and local housing agencies in the UAE to reconsider the designs of vertical public housing to address the identified gaps, specifically in terms of social interaction, high-quality living environments, mixed-use facilities, accessibility, and community involvement in the design process. This can be guided by the various implemented design approaches that have been demonstrated in the global case studies, especially the design solutions that would be more acceptable to the local Emirati citizens. Further research will explore the opinions of the Emirati residents of these recently developed vertical social housing projects about the social sustainability aspects and explore their attitudes towards possible solutions for the defined gaps by utilizing both conventional and modern explorational technologies such as virtual reality.

**Author Contributions:** Conceptualization, O.S.M. and K.G.A.; methodology, O.S.M. and K.G.A.; software, O.S.M.; validation, O.S.M. and K.G.A.; formal analysis, O.S.M.; investigation, O.S.M.; resources, O.S.M. and K.G.A.; data curation, O.S.M. and K.G.A.; writing—original draft preparation, O.S.M.; writing—review and editing, K.G.A.; visualization, O.S.M.; supervision K.G.A.; project administration K.G.A.; funding acquisition, K.G.A. All authors have read and agreed to the published version of the manuscript.

**Funding:** This research was funded by the United Arab Emirates University, through the Strategic Research Program, Grant Code: G00004176.

**Institutional Review Board Statement:** Not applicable.

**Informed Consent Statement:** Not applicable.

**Data Availability Statement:** Not applicable.

**Acknowledgments:** The researchers would like to thank the Emirates Center for Happiness Research and the Research Office at the United Arab Emirates University for supporting this research.

**Conflicts of Interest:** The authors declare no conflict of interest. The funders had no role in the design of the study; in the collection, analysis, or interpretation of data; in the writing of the manuscript; or in the decision to publish the results.

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
