# Peer review of "From Single- to Multi-Family Public Housing: Analyzing Social Sustainability Aspects of Recent Designs in the UAE"

_socsci, doi:10.3390/socsci12090513_

Round 1

Reviewer 1 Report

You should explain why the UAE Federal and local governments "shifted their focus towards providing affordable multi-family housing for your Emerati citizens between the ages 20-30" (Para. 1, p. 11). Was this related to demographic trends?

This study notes that there was no involvement of Emerati residents in the development of the two vertical projects (pp. 9, 11). Given the design gaps identified, have the occupants of these projects complained since their completion about these to the management (if known)? If not, I assume that these might emerge in the recommended future surveys of Emerati Citizen opinions about the design of vertical public housing

Author Response

Comment 1: You should explain why the UAE Federal and local governments "shifted their focus towards providing affordable multi-family housing for your Emerati citizens between the ages 20-30" (Para. 1, p. 11). Was this related to demographic trends?

Response 1: We have added the paragraph in the Introduction section to explain the reason why the UAE Federal and local governments shifted their focus towards providing affordable multi-family housing for your Emerati citizens between the ages 20-30. It reads: “This shift might be attributed to both economic and sustainable development reasons. Economically, developing mid- and high-rise residential blocks would help provide much more numbers of social housing dwellings compared to single family housing. As for sustainable development, such types of vertical social housing help preserve land and conserve resources, especially related to infrastructure and roads”.     

Comment 2: This study notes that there was no involvement of Emerati residents in the development of the two vertical projects (pp. 9, 11). Given the design gaps identified, have the occupants of these projects complained since their completion about these to the management (if known)? If not, I assume that these might emerge in the recommended future surveys of Emerati Citizen opinions about the design of vertical public housing.

Response 2: The two vertical social housing projects have been recently constructed and delivered to the Emirati occupants.Hence, their opinions have not been explored yet. Accordingly, we are going to investigate their opinions in the second face of this study. This has been added to the further research paragraph by the end of the conclusion section.   

Reviewer 2 Report

General assessment

The present paper mobilizes a set of ideas relating to social sustainability aspects that should be taken into consideration when designing multi-family housing public projects in general in the UAE.

The major interest has to do with how critically it explores the relations between social housing, sociodemographic variables, and the cultural context of the UAE in general. But above all, this paper is important because it bridges the gap defining eight principles for social sustainability in vertical housing and then promoting a simple international analysis compared with other countries from different world geographic contexts.

It is well written and structured, and in spite of some theoretical, conceptual and methodological shortcomings, we recommend the article for publication, but only after some changes introduced by the author. The suggested changes, some more profound, others are more superficial, follow the next point with a more specific comment.

Specific assessment

1. The abstract needs to be fully summarized and restructured in order to clearly highlight the paper's objectives and main findings. The author loses a lot of space in the abstract with empirical details that can be summarized so as not to tire the reader and better captivate the reading of the paper.

2. In the introduction, better explain the extent to which public housing in the UAE seeks to achieve the aforementioned United Nations Sustainable Development Goals

3. The introduction is very vague about the intentions and objectives of the paper. Despite the relevance of the key issues raised at the end of this section, these introductory paragraphs can hardly be considered a good introduction. The author should therefore review the construction of the introduction. The introduction it must be an overview of the contents of the research in the paper without going into too much detail. Only a few paragraphs are enough. Briefly describe the importance of the study area. Specify the relevance of the publication of the current paper, ie, explain how this present work contributes to the progress of knowledge in this line of research in urban studies.

4. Lines 133-145: When the author refers to mixed use, it should be noted whether we are talking about social mix or function mix.

5. Lines 148-158: in addition to all aspects of social interaction, the author should also refer to the issue of the primary social network of neighborhood that sometimes promotes strong mutual help; but also to highlight the issue of collective memory and the territorial identity of the neighborhoods that is fostered whenever interaction and social appropriation of space takes place.

6. Some excerpts from the paper need to be more theoretically sustained and coherent or explained based on more facts: line 260 "However, the design of the Diba Al Hosn project lacks social interaction, which is an important aspect of socially sustainable design"; line 267 "Unfortunately, this project does not incorporate parks or green spaces to enhance the quality of the living environment"; line 298 "However, the project was designed without considering any mixed use in the design, resulting in a lack of social interaction within this vertical housing development."  - the author should try to explain these sentences;

7. The limitations of the Diba Al Hosn project in Sharjah Emirate(a) and the Al Ghurfa high-rise social housing project in Al Fujairah Emirate need to be described in more depth. It is not enough to state which aspects of social sustainability are or are not achieved. It is necessary to explain why. Sometimes it is enough to add a sentence in each aspect to make it clearer to the reader.

8. We can also find some limitations in the paper in terms of references, including the lack of standardization and compliance, in its final list. The section of the bibliography in its implementation requires time and detailed attention to certain details. The author must make an effort to maintain the required style and standards of publication in the journal for all references cited and not just some.

Final assessment

The paper presents an important contribution to urban studies, presenting a good empirical demonstration of results and being drafted clearly and objectively. Also reveals a good scientific terminology and vocabulary in the study of public housing and social sustainability.

Despite the requested changes, the author should understand that his/her paper is very good and just sometimes what it takes is to add a small paragraph or phrase on the subjects required, or even change or replace a reference, to significantly improve the work. In short, in spite of the considerations outlined in the field of specific assessment, this review is in favor of publishing the manuscript but with minor modifications, with the expectation that this will surely be an important contribution to the academic and political debates surrounding the public housing, social sustainability and urban regeneration.

Minor editing of English language required

Author Response

Comment 1: The abstract needs to be fully summarized and restructured in order to clearly highlight the paper's objectives and main findings. The author loses a lot of space in the abstract with empirical details that can be summarized so as not to tire the reader and better captivate the reading of the paper.

Response 1: the abstract has been summarized as advised.

Comment 2: In the introduction, better explain the extent to which public housing in the UAE seeks to achieve the aforementioned United Nations Sustainable Development Goals.

Response 2: a sentence and a reference has been added to the Introduction stating that “According to the adopted SGDs agenda by the federal and local governments in the UAE all sectors of urban development including public housing development should comply with this agenda to ultimately achieve the defined goals by 2030 [13]”.

Comment 3: The introduction is very vague about the intentions and objectives of the paper. Despite the relevance of the key issues raised at the end of this section, these introductory paragraphs can hardly be considered a good introduction. The author should therefore review the construction of the introduction. The introduction it must be an overview of the contents of the research in the paper without going into too much detail. Only a few paragraphs are enough. Briefly describe the importance of the study area. Specify the relevance of the publication of the current paper, ie, explain how this present work contributes to the progress of knowledge in this line of research in urban studies.

Response 3: In this research, the Introduction is meant to provide some contextual/historical background about the public housing process and product to provide the international reader with some familiarity of the physical context of public housing in the UAE and its transitional phases. Still, for more clarity about the main objectives of the research a paragraph has been added by the end of the Introduction section as follows “This study is ultimately hoped to shed some light on social sustainability aspects of this new public housing typology especially from social sustainability point view. So, it would contribute to the efforts of achieving SDGs as per the adopted local agendas in the UAE. On the other hand, the study contributes to the current debate about achieving social sustainability in vertical public housing where many efforts in this regard have been discussed globally but with very little attention to such an experience in this part of the world”.

Comment 4: Lines 133-145: When the author refers to mixed use, it should be noted whether we are talking about social mix or function mix.

Response 4: We mean functional mixed use, and this clarification has been added to the related paragraph.

Comment 5: Lines 148-158: in addition to all aspects of social interaction, the author should also refer to the issue of the primary social network of neighborhood that sometimes promotes strong mutual help; but also to highlight the issue of collective memory and the territorial identity of the neighborhoods that is fostered whenever interaction and social appropriation of space takes place.

Response 5: The issue of residents’ identification with the place and the collective sense of belonging among residents have been added to the Social Interaction principle through some references about the social interaction principle on local community levels.  

Comment 6: Some excerpts from the paper need to be more theoretically sustained and coherent or explained based on more facts: line 260 "However, the design of the Diba Al Hosn project lacks social interaction, which is an important aspect of socially sustainable design"; line 267 "Unfortunately, this project does not incorporate parks or green spaces to enhance the quality of the living environment"; line 298 "However, the project was designed without considering any mixed use in the design, resulting in a lack of social interaction within this vertical housing development."  - the author should try to explain these sentences.

Response 6:

For the sentence in Line 260, justification has been added as follows: “the design of Diba Alhosn project is based on separate apartments in each floor with limited collective social activity and social gathering spaces in a way that might undermine social interaction among residents”.

For the sentence in Line 267, justification has been added as follows: “the project design also adopts mainly residential functional use in all typical floors with some limited public uses on the ground floor (reception, parking, shops, and technical services). This means that no or limited recreational spaces are incorporated in the design. Apparently, this would negatively affect the quality of the living environment”.

For the sentence in Line 298, justification has been added as follows: “the project is designed as a residential multi-family block with just some limited technical services in the common spaces that dimensioned the possibility for social interaction except in the main entrance lobby”.

Comment 7: The limitations of the Diba Al Hosn project in Sharjah Emirate and the Al Ghurfa high-rise social housing project in Al Fujairah Emirate need to be described in more depth. It is not enough to state which aspects of social sustainability are or are not achieved. It is necessary to explain why. Sometimes it is enough to add a sentence in each aspect to make it clearer to the reader.

Response 7: Some justification sentences have been added to the analysis as follows. For Diba Alhosn: “It might be argued that most of the aforementioned limitations refer to the consideration on economic efficiency and the desire to achieve sustainability through conserving use of the construction and infrastructure resources.”

For Al Ghurfa project: “Similar to the case of Diba Alhosn and despite noticeable enhancement in some social sustainability aspects in Al Ghurfa project, the limitation of the other social sustainability aspects might be referred to reasons related to economic considerations and resources reservation”.

Comment 8: The author must make an effort to maintain the required style and standards of publication in the journal for all references cited and not just some references.

Response 8: the reference list has been amended to abide by the required style and standards of publication in the journal.

Reviewer 3 Report

The study reveals a solid understanding of the general requirements that have to be met for publishing in a journal nowadays. One of them is the comparative perspective, that asks for any subject matter to be presented as part of an array of examples on a larger international, or regional scale. Here lies both the strenght and the weakness of the article. There's a bit of a rush to put the UAE case alongside examples from "developed countries", which are - of course - more easily accessible, but it also requires, in my opinion, a more in depth analysis of the vast differences in between the local contexts. The authors are bringing some very interesting information about local policies, but this could be supplemented with some other critical reflections on the particularities of the Emirates social protection system, urban planning policies and, not at last, lifestyle and the overwhelming geographical determinations. In this respect, the result is that the case is part of a series of examples that it hardly belongs to, but I do see the value of orienting the theoretical dimension of the study by leaning on the selected cases. Overall, the work is solid and brings in a very interesting description of a place that is under-represented in the general literature on housing by using a method that illustrates most of the research questions that are asked. I would recommend a more detailed description of the social and physical environment that might help an international audience that is unaware of the particularities of the Emirates to place the research in this very genuine context.

Author Response

Comment 1: There's a bit of a rush to put the UAE case alongside examples from "developed countries", which are - of course - more easily accessible, but it also requires, in my opinion, a more in depth analysis of the vast differences in between the local contexts. The authors are bringing some very interesting information about local policies, but this could be supplemented with some other critical reflections on the particularities of the Emirates social protection system, urban planning policies and, not at last, lifestyle and the overwhelming geographical determinations. In this respect, the result is that the case is part of a series of examples that it hardly belongs to, but I do see the value of orienting the theoretical dimension of the study by leaning on the selected cases.

Response 1: In response to this comment, as lifestyles have become somehow ‘glocal’ in nature (i.e merging between local and glocal styles) so it is important to put the analysis in this ‘glocal’ context by revieing some worldwide experiences. Accordingly, the paper is discussing case studies from both developed and developing countries. This includes examples from Netherlands, UK, Denmark and France as cases form the developed counties, and Singapore, China, and Mexico as cases from the developing countries. As for emphasizing locality, some extra background about the public housing process and product as well as the stages of transformations have been added to the Introduction.   

Comment 2: I would recommend a more detailed description of the social and physical environment that might help an international audience that is unaware of the particularities of the Emirates to place the research in this very genuine context.

Response 2: In the introduction we added some background information with related references related to the social and physical contexts of Emiratis public housing including social aspects and the relation between the design of the Emirati house and the local socio culture aspects of Emirati families.